# Protein Misfolding and Aggregation as a Therapeutic Target for Polyglutamine Diseases

**DOI:** 10.3390/brainsci7100128

**Published:** 2017-10-11

**Authors:** Toshihide Takeuchi, Yoshitaka Nagai

**Affiliations:** Department of Neurotherapeutics, Osaka University Graduate School of Medicine, Suita, Osaka 565-0871, Japan

**Keywords:** polyglutamine diseases, Huntington’s disease, conformational change, aggregation, misfolding, β-sheet monomer, therapeutic target, QBP1, molecular chaperone

## Abstract

The polyglutamine (polyQ) diseases, such as Huntington’s disease and several types of spinocerebellar ataxias, are a group of inherited neurodegenerative diseases that are caused by an abnormal expansion of the polyQ tract in disease-causative proteins. Proteins with an abnormally expanded polyQ stretch undergo a conformational transition to β-sheet rich structure, which assemble into insoluble aggregates with β-sheet rich amyloid fibrillar structures and accumulate as inclusion bodies in neurons, eventually leading to neurodegeneration. Since misfolding and aggregation of the expanded polyQ proteins are the most upstream event in the most common pathogenic cascade of the polyQ diseases, they are proposed to be one of the most ideal targets for development of disease-modifying therapies for polyQ diseases. In this review, we summarize the current understanding of the molecular pathogenic mechanisms of the polyQ diseases, and introduce therapeutic approaches targeting misfolding and aggregation of the expanded polyQ proteins, which are not only effective on a wide spectrum of polyQ diseases, but also broadly correct the functional abnormalities of multiple downstream cellular processes affected in the aggregation process of polyQ proteins. We hope that in the near future, effective therapies are developed, to bring hope to many patients suffering from currently intractable polyQ diseases.

## 1. The Polyglutamine Diseases

The polyglutamine (polyQ) diseases are a group of inherited neurodegenerative diseases characterized by a genetic mutation of cytosine-adenine-guanine (CAG) triplet repeat expansion in the coding regions of the disease-causative genes [1,2,3]. The CAG codon encodes the amino acid glutamine (one-letter code, Q), and its expansion in the disease-causative genes, therefore results in the production of mutated proteins with an abnormally expanded polyQ tract. In 1991, Fischbeck and coworkers first reported the disease-associated expansion of CAG repeat in exon 1 of the androgen receptor gene in patients of spinal and bulbar muscular atrophy (SBMA) [4]. Since then, similar genetic mutations of the CAG repeat expansion in the coding regions of genes other than androgen receptor gene have also been found in other inherited neurodegenerative disorders [5,6,7]. So far, nine disorders are reported to belong to this type of diseases, including Huntington’s disease (HD), spinocerebeller ataxia (SCA) types 1, 2, 3, 6, 7 and 17, and dentatorubral pallidoluysian atrophy (DRPLA) (Table 1) [2]. A common pathological feature of the polyQ diseases is progressive degeneration of neurons in the distinct regions of the brain, which causes a variety of neurological and psychiatric symptoms such as cognitive impairment and motor disturbance, depending on the brain regions affected in each disease. Effective therapies to delay or prevent the onset and progression of the polyQ diseases have not yet been established to date.

## 2. Pathogenic Mechanism of Polyglutamine Diseases

### 2.1. A Gain of Toxicity in PolyQ Diseases

The polyQ diseases are inherited in an autosomal-dominant manner, except for SBMA, which is inherited in an X chromosome-linked recessive manner. In the case of HD, patients with homozygous mutations are reported to show apparently no more severe clinical phenotypes than typical patients with heterozygous mutation, suggesting that HD is caused by a true dominant mutation [8,9]. However, in the later study, the disease-associated mutation in a double dose in homozygotes was reported to likely affect the disease progression but not the age of onset [10]. Gene dose effects were also reported in other polyQ diseases including SCA3 and SCA6 [11,12]. These facts indicate that a gain of toxic function mechanism would be involved in the pathogenesis of polyQ diseases. Indeed, loss of the disease-associated genes such as AR and huntingtin, a disease-causative gene of HD, shows no typical disease phenotypes likely observed in the patients of the polyQ diseases, ruling out a loss of function pathway for the disease pathogenesis [13,14]. A study using knock-out mice also demonstrated that heterozygous disruption of Hdh gene, a mouse homologue of human huntingtin gene, did not show apparent disease phenotypes, while homozygous disruption resulted in embryonic lethality, indicating that huntingtin has essential roles in the embryonic development, and loss of huntingtin does not mimic HD neuropathology [15]. Considering these facts, the polyQ diseases are caused mainly by a gain of function mechanism attributed to a single genetic mutation of CAG repeat expansion.

### 2.2. Expansion Mutation of the PolyQ Tract in Pathogenesis

Huntingtin has a polyQ tract at its amino-terminus with a repeat length ranging from 5–35 repeats in normal subjects. In contrast, the polyQ tract of huntingtin in patients of HD is abnormally expanded to longer than 40 repeats [5]. In general, there is a threshold length of polyQ repeats for disease manifestation at around 35 to 40 repeats, and most importantly, longer repeats are associated with earlier, sometimes juvenile, age of onset and increased disease severity [16,17,18,19,20,21,22,23]. Despite shared clinical pathology of progressive loss of neurons among the polyQ diseases, there is no structural homology in primary amino acid sequences and secondary/tertiary structures, as well as no common biological functions among the disease-causative proteins. Abnormal expansion of the polyQ tract is the only characteristic shared among the disease-associated proteins, suggesting the importance of expansion mutation of the polyQ tract in disease pathogenesis. Indeed, studies on animal models using mice [24,25], *Drosophila* [26] and *C. elegans* [27] have demonstrated that expression of the expanded polyQ stretch alone, or artificial proteins fused with an expanded polyQ tract, results in progressive degeneration of neurons and motor disturbance, suggesting that the expanded polyQ tract is sufficient to cause typical phenotypes of the polyQ diseases. The polyQ-dependent pathogenesis has also recently been confirmed in a common marmoset transgenic model of SCA3, which was generated as the first primate model of the polyQ diseases [28]. These facts collectively indicate that the abnormal expansion of the polyQ repeat in disease-causative proteins has a pivotal role in the pathogenic mechanism of the polyQ diseases: recent studies have suggested that repeat RNA transcripts produced from sense/antisense sequences of the polyQ-disease genes, as well as proteins that are unconventionally translated from their transcripts via repeat-associated non-ATG (RAN) translation, also contribute to the pathogenesis of polyQ diseases [29,30,31,32]. Specific functions of each host protein are, therefore, not considered to have a primary role in the pathogenesis of these diseases, although the expansion mutation of the polyQ stretch may affect structure and function of each host protein, which results in aberrant association with key proteins of essential cellular processes, leading to dysfunctions in transcription, proteasomal degradation, synaptic transmission, axonal transport, and Ca^2+^ signaling pathways in the downstream of the pathogenic cascades.

### 2.3. Inclusion Bodies and Aggregates of Proteins with Expanded PolyQ Tracts

In 1997, it was reported that the intranuclear inclusions of the expanded polyQ proteins were formed in the patient brains of the polyQ diseases including HD, SCA3 and DRPLA [33,34,35]. These deposits were also observed in the experimental models, such as cultured cells, *Drosophila*, and mice, in which the expanded polyQ proteins are ectopically expressed [26,36,37,38]. Wanker and coworkers performed ultrastructural analysis of these abnormal structures, and revealed the presence of amyloid-like fibrillar structures, not only in the aggregates formed in vitro, but also in the inclusion bodies in the brains of the transgenic mice models [37]. These fibrillar structures are similar to those observed for scrapie prions and amyloid-β fibrils in Alzheimer’s disease, implying that the polyQ diseases may share pathogenic mechanisms with such amyloid diseases.

While abnormal accumulation of the polyQ proteins such as inclusion bodies is one of the major pathological hallmarks commonly observed in the brains of the polyQ disease patients, the roles of aggregate/inclusion formation in disease pathogenesis have been controversial. Since formation of intranuclear inclusions in a transgenic mouse model of HD is followed by the onset of disease phenotypes [38], and the level of inclusion bodies formed in brains apparently correlates with the severity of polyQ disease phenotypes, they were considered to be responsible for neurodegeneration in the polyQ diseases. A variety of cellular proteins, including molecular chaperones, cytoskeletons, transcriptional factors and proteasomes, are found to be incorporated into the inclusion bodies [39,40,41], implying that sequestration of these cellular proteins in inclusions may cause detrimental effects on a wide range of essential cellular functions, which probably contribute to neuronal dysfunction and eventual loss of neurons in various regions of the brain. However, the discrepancy between inclusion body formation and neurodegeneration has been reported. For example, Li and colleagues examined brain tissues of HD patients at different stages, and found that only 1%–4% of striatal neurons have nuclear aggregates that are immunoreactive with the huntingtin antibody EM48, although the striatum is the most affected region in HD [42,43]. Similar discrepancy was also reported in both patients and a transgenic mouse model of SCA2, where intranuclear inclusions were undetectable while disease phenotypes were exhibited [44]. Furthermore, in the cellular model of HD, Saudou et al. showed that overexpression of mutant huntingtin results in similar levels of inclusions formed in striatal and hippocampal neurons, but causes cell death with different efficiency [45]. These studies suggest that although inclusion bodies are associated with the polyQ diseases, formation of inclusions may not be correlated with the severity of neurodegeneration.

Interestingly, Saudou et al. also reported that expression of a dominant negative mutant of ubiquitin-conjugating enzyme cell division cycle 34 (Cdc34) resulted in a significant decrease in intracellular inclusions formed in the cellular model of HD, but increased cytotoxicity [45]. Muchowski et al. found the essential role of microtubule networks in inclusion body formation in a yeast model of HD, and demonstrated that treatment of yeast with chemicals that disrupt microtubule networks, however, facilitates the polyQ-induced toxicity: perturbation of microtubule-dependent intracellular trafficking results in suppression of inclusion body formation, leading to an increase in the levels of huntingtin protein in a soluble, non-aggregated form [46]. These studies indicate that polyQ-induced cell death is accelerated under certain conditions where inclusion body formation is suppressed. Intracellular inclusions are, indeed, thought to be less toxic structures, as formation of inclusion body might be a cellular protective response to sequester abnormal protein species such as misfolded proteins and aggregates [47,48,49]. In support of this idea, Arrasate et al. demonstrated that the levels of diffused polyQ proteins, rather than those of inclusions, accurately predict an increased risk of death [50]. These studies suggest that it is not inclusion bodies, but other soluble species produced during the aggregation process before inclusion body formation, that likely show higher toxicity within the cells expressing the expanded polyQ proteins.

### 2.4. Abnormal Conformational Changes of Expanded PolyQ Proteins

Since the abnormally expanded polyQ proteins would gain cytotoxicity during the aggregation process, the molecular mechanisms as to how inclusions/aggregates are produced from the monomeric form of the expanded polyQ proteins should be elucidated. Peruz et al. first reported that the chemically synthesized polyQ peptide with a relatively short glutamine repeat (15 repeats of glutamines), under certain conditions, forms aggregates with β-sheet rich structures [51]. Using synthetic polyQ peptides similar to those in the above report but with longer glutamine repeats, Wetzel and coworkers demonstrated that the polyQ peptides gradually undergo a conformational change from a solubilized form with random coil structure to insoluble amyloid-like fibrils with β-sheet structures [52]. Poirier et al. examined the aggregation of huntingtin fragment, and found that intermediate species such as oligomers and protofibrils are produced before amyloid formation of the polyQ proteins, which are structurally similar to those of amyloid-β and α-synuclein formed in Alzheimer’s disease and Parkinson’s disease, implying the pathogenic mechanism shared among the neurodegenerative diseases associated with protein aggregation [53]. In the subsequent studies, various techniques including atomic force microscopy (AFM), electron microscopy, and fluorescence correlation microscopy (FCS) have then been employed, not only to characterize the oligomers of the expanded polyQ proteins, but also to analyze the process of oligomer formation in vitro [54], in cells [55,56] and their extracts [57], and in brain homogenates of model mice [58,59]. We also examined the kinetics of aggregation formation using a model polyQ protein fused with thioredoxin (Thio-polyQ), and confirmed that soluble Thio-polyQ proteins with the expanded polyQ repeat gradually form insoluble aggregates in a time-, concentration-, and repeat length-dependent manner [60]. Importantly, we demonstrated that conformational transition of Thio-polyQ protein to a β-sheet rich structure occurs in its monomeric state, followed by assembly into oligomers and insoluble amyloid fibrils [61]. These studies indicate that proteins with the expanded polyQ tract undergo a conformational transition from the native conformer to the β-sheet rich structure in a monomeric state, which assembles into oligomers and insoluble aggregates with amyloid fibrillar structures, potentially leading to accumulation as intracellular inclusions (Figure 1).

Accumulating evidence suggest that abnormal intermediate species in a soluble state such as oligomeric intermediates and even misfolded monomers of the expanded polyQ proteins could be more toxic to neurons compared with insoluble aggregates/inclusion bodies [62]. Kayed et al. reported that treatment of cells with soluble oligomers of the polyQ protein results in significant cell death, which is inhibited by oligomer-specific antibodies, suggesting the potential cytotoxicity of the soluble intermediate species formed during the aggregation process [63]. This is in agreement with the results of our studies performed using cultured cells, where we employed two different techniques, FCS and fluorescence resonance energy transfer (FRET) confocal microscopy, that enable the presence and dynamics of intermediate soluble oligomers of the polyQ proteins in cells to be analyzed [55,56], and demonstrated that cells with soluble oligomers died faster than those with inclusions. In addition, we isolated in vitro the soluble oligomers and misfolded monomers of the model polyQ protein Thio-polyQ, and tested their cytotoxicity by microinjecting these species to the cultured cells, and found that a monomeric conformer with a β-sheet structure, as well as oligomers, showed significant cytotoxicity [61]. These studies collectively indicate that cytotoxicity of the expanded polyQ proteins would arise from an abnormal conformational transition to β-sheet rich structure and oligomer formation, which occurs during the aggregation process of the polyQ proteins before inclusion body formation.

## 3. Therapeutic Approaches for Polyglutamine Diseases: Targeting Misfolding and Aggregation of Expanded Polyglutamine Proteins

As discussed above, the abnormally expanded polyQ tract is structurally unstable, and likely undergoes a conformational change to the misfolded states, resulting in an assembly of its host protein into insoluble aggregates with β-sheet rich amyloid fibrillar structures. The disease-causative proteins with abnormally expanded polyQ tracts gain cytotoxicity during the aggregation process, although it is still unclear which intermediate species appearing via this process are responsible for the pathogenesis of the polyQ diseases. Since misfolding and aggregation of the expanded polyQ tract are thought to be an initial event in the common pathogenic cascade of the polyQ diseases, suppression of the polyQ aggregation is expected not only to operate on a wide spectrum of the polyQ diseases, but also to broadly correct the functional abnormalities of multiple downstream cellular processes affected in the aggregation process of the polyQ proteins (Figure 1). Taking these advantages, suppression of misfolding and aggregation of the polyQ proteins has been extensively studied as an ideal therapeutic approach for development of disease-modifying therapies for the polyQ diseases. In the following subsections, we focus on two approaches targeting to the polyQ aggregation: one approach is to develop potent inhibitors such as small chemical compounds and short peptides that are designed or selected to bind specifically to the expanded polyQ tract, and suppress the aggregation process of the polyQ proteins. The other approach is to activate cellular protective systems that prevent aggregate formation and accumulation of the misfolded proteins. Both approaches are effective not only to suppress the aggregation process of the expanded polyQ proteins, but also to suppress disease progression and phenotypes in the animal models, showing the effectiveness of this therapeutic target for drug development.

### 3.1. Suppression of Polyglutamine Aggregation by Inhibitor Peptides and Chemicals

It has been reported that an antibody against the polyQ tract, 1C2, binds preferentially to longer polyQ repeats rather than shorter ones [64], implying that the polyQ tract would have different structures depending on its repeat length [65]. This raises the possibility that molecules that selectively bind to the expanded polyQ tract could be promising drug candidates, as they are expected to stabilize the unique structure and to interfere with the aggregation processes of the polyQ proteins. Based on this idea, we performed a phage display screening to develop short peptides that would bind to the expanded polyQ proteins with high affinities, and identified several peptides that preferentially bind to the abnormally expanded polyQ stretch [60]. Among them, polyQ binding peptide 1 (QBP1) effectively suppresses aggregation formation of the expanded polyQ proteins in vitro, and suppresses polyQ-induced cytotoxicity in cultured cells. Importantly, QBP1 has been shown to inhibit β-sheet conformational transition of the polyQ protein monomer as well as oligomer formation, which occurs at an early stage of the aggregation cascade of the expanded polyQ proteins, and thus, would be one of the most promising therapeutic targets in the aggregation process [55,61]. Expression of QBP1 effectively suppresses polyQ inclusions as well as polyQ-induced neurodegeneration in *Drosophila*, supporting its therapeutic potential for the polyQ diseases [66]. However, QBP1 shows limited therapeutic effects on the mouse models of HD upon its peripheral administration [67], probably due to the poor efficiency of peptide-based drugs to pass through the blood-brain barrier (BBB). These results suggest that QBP1 is one of the ideal drug candidates that suppress the early events responsible for the polyQ disease pathogenesis, but further structural optimization would be necessary to overcome the delivery efficiency to the brain.

In parallel with our study, the antibody 1C2 was also shown to inhibit fibril formation of the expanded polyQ proteins in vitro in a dose-dependent manner [68]. Similarly, single chain antibodies, i.e., intrabodies, that have high affinity to the abnormally expanded polyQ proteins, have been reported to suppress formation of the polyQ aggregation and the polyQ-induced toxicity in the polyQ disease models of cells [69,70,71], *Drosophila* [72], and mice [73]. 

Chen et al. also performed combinatorial screening to search for potential inhibitors of polyQ aggregation using a combinatorial library consisting of peptoids, which are oligomers of N-substituted glycines, and have superior advantages in stability to protease degradation, cell permeability, and structural diversity. From 60,000 unique peptoid library, they isolated a peptoid HQP09 (Huntingtin poly-Q binding Peptoid 09), which binds with high specificity to the expanded polyQ proteins of huntingtin and ataxin-3, a causative protein of SCA3 [74]. HQP09 effectively suppress polyQ aggregation in vitro, reduced cytotoxicity in primary cultured neurons and decreased polyQ inclusion bodies in a mouse model of HD upon its intracerebroventricular injection. Importantly, they successfully identified the pharmacophore of HQP09 based on a structure-activity relationship study, and developed the minimal derivative peptoid HQP09-9 (4-mer, MW = 585) without significant loss of activity. Although subcutaneous injection of HQP09-9 failed to exert therapeutic effects on a mouse model probably due to poor BBB permeability, this could be a promising lead compound for the development of drugs against a broad spectrum of the polyQ diseases.

Small chemical compounds that have inhibitory activities for polyQ aggregation have also been developed. Wanker and coworkers first reported that several compounds including Congo red effectively suppress the polyQ aggregation in vitro [68]. Congo red was shown to reduce polyQ inclusions and improve motor deficits and survival in the model mice of HD via systemic administration [75], although the improvement has not been reproduced by other groups, probably due to the inability of this compound to cross the BBB [76]. Wanker’s group also developed an automated filter retardation assay and performed high-throughput screening using a large-scale chemical library (~184,000 compounds) to identify compounds that prevent aggregation formation of the expanded polyQ proteins [77]. Using this method, they identified about 300 chemical compounds that suppress the polyQ aggregation in a dose-dependent manner. Among them, benzothiazoles were thought to be quite promising, as benzothiazole and its related structures appeared commonly in 25 hit compounds, which efficiently suppressed aggregation formation of the polyQ proteins not only in vitro, but also in cultured cells. However, therapeutic effects of PGL-135, the most promising benzothiazole compound, was not able to be confirmed using mouse models, as this compound was metabolically unstable with an extremely short half-life after intraperitoneal injection, although this compound was able to cross the BBB [78]. Wanker’s group also found that epigallocatechingallate (EGCG), a green tea polyphenol, modulates misfolding and oligomerization of the expanded polyQ proteins, resulting in efficient suppression of polyQ aggregation in vitro [79]. EGCG suppressed aggregation formation of the polyQ proteins and polyQ-induced cytotoxicity in HD models of yeast and *Drosophila* [79], and the SCA3 model of *C. elegans* [80,81]. Importantly, EGCG was shown to redirect amyloid-like fibril formation of the polyQ proteins toward less-toxic off-pathway species that are poor in β-sheet structures with increasing solubility [80]. We have also performed chemical library screening to find compounds that suppress aggregation formation of the polyQ proteins using model polyQ protein Thio-polyQ. From 46,000 compounds, we identified about 100 compounds that potently inhibit the polyQ aggregation in vitro (unpublished), some of which have currently been tested for their therapeutic effects using the polyQ models of *Drosophila* and mice.

### 3.2. Suppression of Polyglutamine Aggregation by Activation of Cellular Proteostasis Networks

Proteins have essential roles in cell survival as one of the major components of living organisms, and therefore quality control of cellular protein production, folding, and degradation must be tightly regulated. To prevent aberrant accumulation of misfolded proteins that would be otherwise toxic, cells have a highly conserved and integrated protective system that maintains cellular protein homeostasis (proteostasis), which includes molecular chaperones, autophagy, and the ubiquitin-proteasome system (UPS). Among them, molecular chaperones have central roles in maintaining cellular proteostasis that assist refolding of misfolded proteins, and mediate degradation through autophagy and proteasome machinery [82]. Since the polyQ diseases are caused by misfolding and aggregation of the disease-associated proteins, activation of cellular protective mechanisms that maintain proteostasis including molecular chaperones is expected to be one of the therapeutic approaches used to treat polyQ diseases.

In 1998, Commings et al. found that overexpression of human DnaJ homolog 2 (Hdj-2), one of Heat shock protein 40 (Hsp40) family proteins, in the cellular model of SCA1 resulted in a significant reduction in inclusion body formation of ataxin-1, a disease-causative protein of SCA1 [39]. Subsequently, several groups revealed that overexpression of molecular chaperones including Hsp70 and Hsp40 in animal models such as *Drosophila* and mice expressing the expanded polyQ proteins, led to the suppression of inclusion formation, as well as to improvement of the typical disease-associated phenotypes, including motor disturbance [83,84]. Moreover, it has been reported that the carboxy terminus of Hsc70-interacting protein (CHIP), a co-chaperone of Hsp70 that works as a E3 ubiquitin ligase, also suppresses polyQ aggregation and improves viability in the cells expressing polyQ proteins [85,86], and suppresses neurodegeneration in animal models of SCA1 and SBMA [87,88]. These reports strongly suggest that the activation of chaperone functions is indeed an effective approach to develop molecular therapies for the polyQ diseases. Other molecular chaperones including Hsp84, Hsp105, and small Hsps such as HspB1 (Hsp27), HspB7, and HspB8 (Hsp22), have also been reported to suppress polyQ-induced cytotoxicity in the cell culture models [89,90,91,92,93].

Heat shock factor 1 (HSF1) is a transcriptional factor that regulates the transcription of most of heat shock proteins, including Hsp70, Hsp90 and Hsp40 [94]. Since transcriptional activity of HSF1 is negatively regulated by Hsp90, inhibition of Hsp90 is expected to activate HSF1 function, leading to the induction of various heat shock proteins. Sittler et al. found that treatment of a Hsp90 inhibitor geldanamycin with the polyQ expressing cells led to induction of chaperones such as Hsp70, Hsp90, and Hsp40, resulting in significant inhibition of aggregation formation of the polyQ proteins, demonstrating for the first time that pharmacological induction of multiple chaperones is an effective therapeutic approach for drug development of the polyQ diseases [95]. Radicicol, a fungal macrocyclic antibiotic that induces heat shock chaperones by a mechanism similar to that of geldanamycin, has also been shown to suppress polyQ aggregation in slice cultures from HD model mice (R6/2) [96]. The other Hsp90 inhibitor, dimethylaminoethylamino-17-demethoxy-geldanamycin (17-DMAG), is shown to reduce nuclear accumulation of mutant androgen receptor (AR), and ameliorate motor impairment in the transgenic mouse model of SBMA by oral administration, although the therapeutic outcomes are considered to be attributed to enhanced degradation of mutant AR through ubiquitin-proteasome system, rather than by chaperone induction: Hsp90 inhibitors enhance the proteosomal degradation of Hsp90 client proteins [97]. In this line of research, Katsuno et al. demonstrated that oral administration of geranylgeranylacetone, a compound that potently induces chaperones, into SBMA transgenic mice, resulted in the up-regulation of various heat shock proteins in the central nervous system, leading to suppression of aberrant nuclear accumulation of the disease-associated androgen receptor and to amelioration of the polyQ-dependent neuromuscular phenotypes [98]. We also tested a series of chaperone-inducing compounds in the fly models of the polyQ diseases, and found that 17-(allylamino)-17-demethoxygeldanamycin (17-AAG), a less toxic derivative of geldanamycin, has a high potency to suppress inclusion body formation as well as to inhibit progressive neuronal loss [99].

To explore the possibility of gene therapy using molecular chaperones for treatment of the polyQ diseases, we introduced Hsp40 gene using the adeno-associated virus (AAV) in the brain of the mouse model of HD. We found that viral vector-mediated overexpression of Hsp40 in mouse brain results in improvement of motor disturbance and survival, as well as a significant reduction in inclusion body formation of polyQ proteins, demonstrating the effectiveness of a gene therapy approach using molecular chaperones for polyQ diseases [100]. Interestingly, we found that, in the brains of the HD mice that were injected with the Hsp40-expressing AAV, inclusion body formation was broadly suppressed, not only in virus-infected cells, but also in the non-infected cells, suggesting the non-cell autonomous therapeutic effects of Hsp40 in vivo. Using cellular and *Drosophila* models of polyQ diseases, we examined the molecular basis underlying this non-cell autonomous effect of Hsp40, and found that molecular chaperones including Hsp40 and Hsp70 are secreted via exosomes, one of the extracellular vesicles, and transmitted to the other cells, where they suppress aggregation formation of the polyQ proteins [101]. These findings indicate that molecular chaperones function not only in a cell-autonomous manner, but also in a non-cell autonomous manner at the multicellular organismal levels, both of which contribute to maintenance of proteostasis. Thus, activation of molecular chaperones is expected to be a highly effective approach for treatment of polyQ diseases.

It has also been reported that activation of the cellular degradation systems such as autophagy and proteasome systems is effective for the treatment of polyQ diseases [102,103], as they can facilitate degradation of the expanded polyQ proteins and suppress polyQ-induced toxicity [104,105,106]. Rubinsztein and colleagues demonstrated that rapamycin, a specific mTOR inhibitor that potently induces autophagy, suppressed polyQ aggregation and polyQ-induced cell death in cell culture models of the polyQ diseases, and suppressed progressive neurodegeneration in fly models [107]. They also showed that intraperitoneal injection of CCI-779, a rapamycin analog, reduced aggregation formation and improved disease phenotypes in mouse models of HD and SCA3 [107,108]. Trehalose, a natural disaccharide with chemical chaperone activity that binds to unfolded proteins and stabilizes their structure, has been reported to inhibit polyQ aggregation in vitro, and effectively improves disease phenotypes in HD model mice by oral administration [109]. Interestingly, it has been found that trehalose can also work as a potent activator of autophagy, and facilitates clearance of mutant huntingtin [110] and other aggregation-prone proteins associated with the neurodegenerative diseases [110,111,112,113]. Considering that trehalose has reduced toxicity and the superior water solubility, coupled with its dual functions as a chemical chaperone and an autophagy inducer, trehalose is a promising drug candidate, not only for polyQ diseases, but also for other neurodegenerative diseases. Shoji-Kawata et al. found that a peptide derived from the autophagy-related protein Beclin1 is a potent inducer of autophagy, and decreases the accumulation of polyQ proteins in the cellular disease models [114]. Paeoniflorin, a major component of Paeonia plants, has been also shown to facilitate clearance of pathogenic AR by enhancing both autophagy and UPS, resulting in therapeutic improvement in behavioral and pathological impairments in SBMA model mice [115]. These results indicate that the therapeutic approach focusing on degradation of expanded polyQ proteins through induction of autophagy is also highly effective for treatment of polyQ diseases. However, due to the non-selective nature of autophagy, activation of autophagy results not only in degradation of the disease-associated polyQ proteins, but also in sequestration of other cellular proteins, which might lead to detrimental effects on cellular activity and survival. To accomplish selective degradation of the expanded polyQ proteins without affecting levels of other proteins, Bauer et al. designed an adaptor peptide comprising QBP1 for polyQ binding and Hsc70-binding motifs, which is expected to specifically bind to proteins with the expanded polyQ tract and direct them to lysosomes through chaperone-mediated autophagy [116]. Expression of this adaptor peptide resulted in selective degradation of the expanded polyQ proteins in cell culture models of HD, and reduced polyQ aggregates and ameliorated the disease phenotypes in the HD mice, indicating the effectiveness of induction of selective autophagy on therapies for polyQ diseases.

## 4. Gene Silencing: An Emerging Approach Targeting Upstream of Pathological Protein Accumulation

Currently, gene silencing strategies have attracted much attention as a promising therapeutic approach for dominantly inherited neurodegenerative diseases including polyQ diseases [117,118]. Yamamoto et al. generated a conditional model mice where expression of the expanded polyQ protein could be regulated through the control of a tetracyclin-based regulatory system, and demonstrated that turning off the expression of the polyQ protein in the symptomatic mice reduces inclusion bodies and ameliorates motor disturbance [119]. This result indicates that neuropathology and disease phenotypes of the polyQ diseases can be reversed, at least at the early stage of disease progression, raising the possibility that reducing the intracellular levels of aberrant polyQ proteins would be a promising approach for treatment of the polyQ diseases. Indeed, transcriptional suppression of gene expression of the disease-causative proteins by the use of endogenous mRNA degradation systems such as RNA interference (RNAi) has been shown to be effective for reducing polyQ aggregation/inclusion and improving disease phenotypes [120,121,122]. So far, antisense oligonucleotides (ASO), short interfering RNAs (siRNA), and short hairpin RNAs (shRNA) have been employed to decrease the level of the disease-associated proteins of the polyQ diseases, including HD, SBMA, and several types of SCAs, and thus, to improve phenotypes of animal models for such diseases [123,124,125,126,127]. In addition, several miRNAs have been identified that are able to reduce the expression level of the polyQ proteins by directly binding to 3’ untranslated regions of their mRNAs [128,129,130], or by indirect mechanisms [131,132].

Despite recent progress in gene silencing strategies for treatment of the polyQ diseases, there are still some challenges to be overcome for practical use. One of these challenges, especially in comparison with the strategy targeting misfolded proteins, includes the technical difficulty in targeting only the mutant alleles without affecting the normal allele. Therefore, most gene silencing strategies have utilized a ‘partial reduction’ approach, in which mutant and normal alleles are both targeted in a non-specific manner, and gene transcription in both alleles are not completely, but partially suppressed. Because host proteins harboring the polyQ tract should possess some functional roles in cells, as exemplified by huntingtin's essential role in development [15], lack of allele specificity in gene suppression possibly causes detrimental effects on normal cellular functions. Methods to allow allele-specific silencing by targeting disease-linked single-nucleotide polymorphisms (SNPs) [133,134,135] and specific conformations formed in abnormally expanded CAG repeats [136] have recently been tested, but these are still being developed (see review: [137]). Although other issues including brain delivery of nucleic acids and off-target effects, which are general issues on nucleic acid therapeutics, also remain to be solved for clinical application, gene silencing strategies are expected to be a promising approach for polyQ diseases; indeed, an ASO that targets the huntingtin gene has been clinically tested in Phase 1/2 studies in early stage patients of HD, initiated from July 2015 by Ionis Pharmaceuticals in collaboration with Roche.

## 5. Future Perspectives

Here we reviewed the current understanding of molecular pathogenic mechanisms of the polyQ diseases, and introduced therapeutic approaches that have been developed, focusing on the pathomechanisms of the diseases. As discussed, polyQ proteins are thought to change their conformation to β-sheet rich structures in a monomeric state, which initiates an aggregation cascade to assemble into soluble oligomers and aggregates, eventually leading to accumulation as inclusion bodies. Several key questions however, still remain elusive. Which intermediate species are responsible for the pathogenesis of polyQ diseases, and how do these abnormal species gain cytotoxicity? How does conformational transition into a β-sheet structure at the monomer level lead to formation of oligomers and aggregates with β-sheet-dominant structures? Is there any possibility of conformational propagation of the polyQ proteins, such as scrapie prion propagation, where a misfolded monomer with β-sheet structure would induce conformational transition of other natively-folded polyQ proteins? Further elucidation of polyQ aggregation and its cytotoxicity not only would lead to better understanding of the molecular mechanisms underlying polyQ disease pathogenesis, but would also provide an important insight into therapeutic targets effective for treating these diseases.

Although detailed mechanisms on polyQ aggregation are yet to be elucidated, it is clear that the early events during the aggregation process, such as abnormal conformational transition to β-sheet structure and oligomer formation, are at this moment the most ideal therapeutic target for development of the disease-modifying therapy of the polyQ diseases. Our results show that QBP1 recognizes and inhibits β-sheet transition of a polyQ protein monomer, indicating that QBP1 would be a promising seed peptide targeting the most upstream event in the aggregation cascade. Peripheral administration of QBP1 into the mouse model of the polyQ disease, however, resulted in limited therapeutic effects, which is probably attributed to poor efficiency for passing through the BBB. Considering the fact that HQP09-9, a peptoid-based drug candidate screened by Chen et al., similarly failed to show the expected therapeutic effects on the polyQ mouse model by subcutaneous injection, brain delivery methods that enable the improvement of the BBB-permeability of drug candidates, should be developed as a top priority issue. Based on structure-function relationship studies [138,139], together with the solution structure of polyQ proteins determined by NMR measurement [140], we are currently working on sequence optimization and derivatization of QBP1, as well as development of its small chemical analogues that are expected to efficiently pass through the BBB. In addition, brain delivery vectors that allow cargo molecules to efficiently translocate across the BBB are also being developed, which will be reported in due course.

So far, several disease-modifying therapies for the polyQ diseases have been clinically tested. One of the representative trials includes leuprorelin, a luteinizing hormone-releasing hormone (LHRH) peptide agonist, which reduces testosterone-dependent AR accumulation in the nucleus and is expected to be effective for SBMA treatment. Subcutaneous injection of leuprorelin in a transgenic mouse model of SBMA decreased nuclear accumulation of mutant AR in muscle and spinal cord improved disease phenotypes such as motor dysfunction, and extended life span [141]. However, a series of clinical trials of a randomized, placebo-controlled study for 48 weeks in SBMA patients showed no significant improvement in motor dysfunction, despite limited outcomes including suppression of AR accumulation and reduced serum level of testosterone [142,143]. Similarly, 5α-reductase inhibitor dutasteride, which is also expected to decrease AR toxicity, has been evaluated clinically in SBMA patients for 24 months, but showed no significant improvement in treatment of SBMA [144]. Although both trials have failed to prove clinical efficacy, subgroup analysis revealed that leuprorelin shows improvement of swallowing function in the early phase patients of SBMA with disease duration less than 10 years, indicating that disease duration is important for clinical evaluation of therapeutics for these diseases. Indeed, long-term treatment with this agent appears to delay functional decline, and shows effectiveness for patients [145]. Leuprorelin has very recently been approved for suppression of disease progression for SBMA patients in Japan.

The other important issue is clinical evaluation of the efficacy of newly developed drug candidates in human patients. Because neurological and psychiatric symptoms of polyQ diseases gradually progress over several years, it would be quite difficult to evaluate the therapeutic outcome of drug candidates during a short period of clinical study. This fact strongly indicates that not only effective drugs, but also disease-linked biomarkers that would faithfully reflect the severity and progression of the disease phenotypes with high sensitivity, are necessary for the development of disease-modifying therapies for polyQ diseases. Much effort has been made to identify proteins and RNAs that can be utilized as biomarkers, by examining body fluids such as cerebrospinal fluid (CSF) and blood from patients, as well as animal models of the polyQ diseases. We hope that in the near future, therapeutic approaches that are widely effective against polyQ diseases are developed, and to bring hope to many patients suffering from the currently intractable polyQ diseases.

## Figures and Tables

**Figure 1 brainsci-07-00128-f001:**
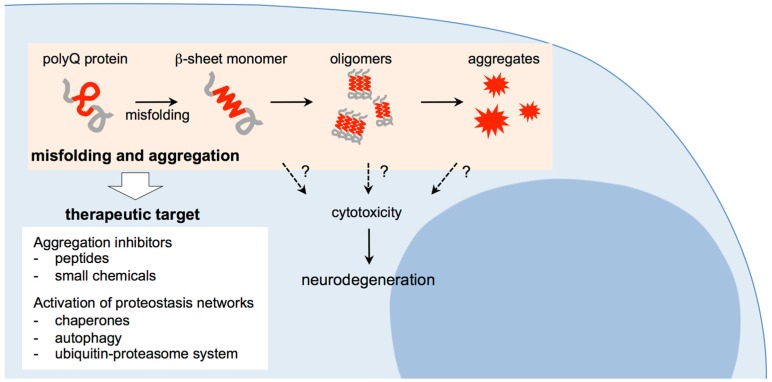
Proposed aggregation cascade of the expanded polyglutamine (polyQ) proteins and potential therapeutic targets for polyQ diseases.

**Table 1 brainsci-07-00128-t001:** The polyglutamine diseases; CAG: cytosine-adenine-guanine.

Disease	Gene	CAG Repeat	
		Normal	Disease
Spinal and bulbar muscular atrophy (SBMA)	androgen receptor	9–36	38–65
Huntington’s disease (HD)	huntingtin	6–35	36–180
Spinocerebeller ataxia type 1 (SCA1)	ataxin-1	6–39	39–83
Spinocerebeller ataxia type 2 (SCA2)	ataxin-2	14–32	32–200
Spinocerebeller ataxia type 3 (SCA3)	ataxin-3	12–41	55–84
Spinocerebeller ataxia type 6 (SCA6)	α1A calcium channel	4–19	20–33
Spinocerebeller ataxia type 7 (SCA7)	ataxin-7	4–35	37–306
Spinocerebeller ataxia type 17 (SCA17)	TATA-binding protein	25–44	46–63
Dentatorubral pallidoluysian atrophy (DRPLA)	atrophin-1	6–36	49–88

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
