# Peer review of "Protein Misfolding and Aggregation as a Therapeutic Target for Polyglutamine Diseases"

_brainsci, 2017, doi:10.3390/brainsci7100128_

Round 1
Reviewer 1 Report
The Authors review the role of protein misfolding and aggregation in polyQ diseases.
The theme is interesting, also because the protein quality control is emerging as a crucial therapeutic target for these diseases. However the present review is not updated with the most recent works, as shown in the bibliography, resulting incomplete. The authors need to update the review.
Moreover, it seems to be not balanced, in fact the review is mainly focused on HD and only marginally to the other polyQ diseases, it requires to enlarge the part regarding the other diseases.
In the paragraph 3.1, it seems to be self-referential, and very detailed in the experiment despite the other paragraphs that are more essential in the description. this paragraph needs to be revised.
3.2 is the paragraph that required a great revision, because it might be interesting for therapy, but it is very incomplete.
There are several chaperones (and protein related to chaperones) that seem to modulate polyQ aggregation but are not described here, as Hsp22, and CHIP for example. I suggest to deepen this aspect.
The description of compounds capable to induce either Hsp expression or to modulate the proteostasis is very poor. The authors have to improve these parte and to include the compounds found effective in polyQ diseases and not described in the text, as radicicol, trehalose (and therapies analogues melibiose and lactulose), 17-DMAG, paeniflorin and other tested in the last years.
in the paragraph 4, the recent works obtained in SBMA [with ASO (by group of A. Lieberman) and with miRNA (by group of K. Fishbeck and G. Sobue)] and in SCA3 (miRNA by group of Jiang) are not cited in the text. Moreover, there is no citation about chemically-modified single stranded siRNA.
To be accepted for publication, this review needs a major revision to originally differ from the others already published in the field.
Author Response
Response to Reviewer #1:
The Authors review the role of protein misfolding and aggregation in polyQ diseases. The theme is interesting, also because the protein quality control is emerging as a crucial therapeutic target for these diseases. However the present review is not updated with the most recent works, as shown in the bibliography, resulting incomplete. The authors need to update the review. Moreover, it seems to be not balanced, in fact the review is mainly focused on HD and only marginally to the other polyQ diseases, it requires to enlarge the part regarding the other diseases.
We are grateful to the reviewer for the insightful comments and helpful suggestions, which were very helpful for us to improve the manuscript. We have taken all your comments and suggestions into account in revising our manuscript, and our detailed responses to each comment are as follows:
Comment 1:
In the paragraph 3.1, it seems to be self-referential, and very detailed in the experiment despite the other paragraphs that are more essential in the description. this paragraph needs to be revised.
Response 1:
We thank the reviewer for this comment. According to the reviewer’s suggestion, we shortened the first paragraph of section 3.1 that is related to our previous work.
Comment 2:
3.2 is the paragraph that required a great revision, because it might be interesting for therapy, but it is very incomplete. There are several chaperones (and protein related to chaperones) that seem to modulate polyQ aggregation but are not described here, as Hsp22, and CHIP for example. I suggest to deepen this aspect.
Response 2:
We thank the reviewer for this important comment. Because chaperones and other proteins that can modulate polyQ aggregation are quite important in regard to development of disease-modifying therapies for the polyQ diseases, we totally agree with what the reviewer pointed out. Therefore, we added a sentence describing the role of CHIP in polyQ suppression, as well as references related to CHIP, in the second paragraph of section 3.2 (page 7, line 302-305). According to the reviewer’s suggestion, small heat shock proteins including HspB8 and others were also added in the last part of the second paragraph of the same section (page 7, line 307-309) in the revised manuscript.
Comment 3:
The description of compounds capable to induce either Hsp expression or to modulate the proteostasis is very poor. The authors have to improve these parte and to include the compounds found effective in polyQ diseases and not described in the text, as radicicol, trehalose (and therapies analogues melibiose and lactulose), 17-DMAG, paeniflorin and other tested in the last years.
Response 3:
We thank the reviewer for this comment. According to the reviewer’s comment, we added sentences describing radicicol and 17-DMAG, and their references, in the third paragraph of section 3.2 (page 8, line 317-324). Furthermore, trehalose and paeoniflorin were also added in the last paragraph of the same section (page 8, line 358-page 9, line 366, and page 9, line 368-371, respectively).
Comment 4:
in the paragraph 4, the recent works obtained in SBMA [with ASO (by group of A. Lieberman) and with miRNA (by group of K. Fishbeck and G. Sobue)] and in SCA3 (miRNA by group of Jiang) are not cited in the text. Moreover, there is no citation about chemically-modified single stranded siRNA.
Response 4:
We thank the reviewer for this important comment. Accordingly, we added references related to the gene silencing approaches for the polyQ disease therapy in the revised manuscript (page 9, line 401-403). In addition, because of the growing volume of the paragraph describing the gene silencing approach, we made a new section for this topic “4. Gene silencing: an emerging approach targeting upstream of pathological protein accumulation,” and most of sentences describing gene silencing approach that was originally in the last section “Future perspective” were moved to the new section.

Reviewer 2 Report
The manuscript by Takeuchi and Nagai reviews promising therapeutic approach for polyglutamine diseases, in which toxic protein is the target.
Generally, the manuscript is well written and covers the reviewed area quite well. I have some comments which could help to refer to current advances in the field.
- Some sentences are too long, for example: page 2, lines 72-76
- Page 2, line 72: References 16 and 17 are related to SBMA, whereas the sentence has a general context for polyQ diseases
- Page 3, line 78: Please check the references here. 18 might not be a good example here, as polyQ tract is expressed in the context of exon1 of HTT – a fragment which actually is present in cells after proteolytic cleavage of huntingtin
- Possible contribution of toxicity of mutant RNA could be briefly mentioned in the description of pathogenesis of polyQ diseases
- The same conclusion is given on page 4, lines 135-137 and page 5, lines 168-170.
- As the review is focused on strategies targeting protein misfolding and aggregation, it would be good to point out the advantages of these approaches, especially in comparison to gene silencing strategies which are mentioned.
- „Future perspective” section: in the reference to allele specificity in gene silencing strategies, two approaches successfully tested on animals can be mentioned: SNP site-targeting and CAG repeat-targeting.
- Issues of potential delivery of potential therapeutics to humans could be discussed.
- Also it would be valuable to mention about clinical trials with humans ongoing, related to polyQ disease-modifying approaches.
Author Response
Response to Reviewer #2:
The manuscript by Takeuchi and Nagai reviews promising therapeutic approach for polyglutamine diseases, in which toxic protein is the target. Generally, the manuscript is well written and covers the reviewed area quite well. I have some comments which could help to refer to current advances in the field.
We are grateful to the reviewer for the insightful comments and helpful suggestions, which were very helpful for us to improve the manuscript. We have taken all your comments and suggestions into account in revising our manuscript, and our detailed responses to each comment are as follows:
Comment 1:
Some sentences are too long, for example: page 2, lines 72-76
Response 1:
We thank the reviewer for this helpful comment. We checked the length of each sentence throughout the manuscript, and tried to shorten the sentences that are longer than 5 lines, including the one that the reviewer pointed.
Comment 2:
Page 2, line 72: References 16 and 17 are related to SBMA, whereas the sentence has a general context for polyQ diseases
Response 2:
We thank the reviewer for this important comment. We added some references that are related to other PolyQ diseases, including HD, DRPLA, SCA1 and SCA3, as Ref.18-23, in addition to those of SBMA (page 2, line 75).
Comment 3:
Page 3, line 78: Please check the references here. 18 might not be a good example here, as polyQ tract is expressed in the context of exon1 of HTT – a fragment which actually is present in cells after proteolytic cleavage of huntingtin
Response 3:
Thank the reviewer for pointing this. We removed Ref.18 in the revised manuscript (page 3, line 81).
Comment 4:
Possible contribution of toxicity of mutant RNA could be briefly mentioned in the description of pathogenesis of polyQ diseases
Response 4:
We thank the reviewer for this important comment. The repeat RNA transcripts produced from the polyQ disease genes and peptides that are translated via repeat-associated non-ATG translation (RAN translation) have recently been reported to contribute to the pathogenic mechanisms of the polyQ disease. We added sentences describing RNA toxicity and references in section 2.2 (page 3, line 88-92) in the revised manuscript.
Comment 5:
The same conclusion is given on page 4, lines 135-137 and page 5, lines 168-170.
Response 5:
We thank the reviewer for pointing this. Although these sentences both describe the possible existence of the polyQ species more toxic to neurons compared with large aggregates/inclusion bodies, the latter sentence particularly highlights oligomers and misfolded monomers of polyQ proteins as more toxic species to neurons, which is quite important for understanding how and when polyQ proteins exert cytotoxicity in their aggregation process. For the above reason, no change was made in the revised manuscript.
Comment 6:
As the review is focused on strategies targeting protein misfolding and aggregation, it would be good to point out the advantages of these approaches, especially in comparison to gene silencing strategies which are mentioned.
Response 6:
We thank the reviewer for the important comments. One of the major advantages of the strategies targeting protein misfolding and aggregation is that these approaches target an upstream event in the common pathogenic cascade of the polyQ diseases, so that they are expected to be effective on a wide spectrum of these diseases, as well as to broadly correct the functional abnormalities of downstream cellular processes affected by the polyQ aggregation, which are described in the first paragraph of section 3 (page 5, line 201-206). In addition, according to the reviewer’s comments, current issues on gene silencing approaches including technical difficulty in targeting only the mutant allele without affecting the normal allele was described in the second paragraph of the new section 4 in the revised manuscript (page 9, line 404-page 10, line 420).
Comment 7:
“Future perspective” section: in the reference to allele specificity in gene silencing strategies, two approaches successfully tested on animals can be mentioned: SNP site-targeting and CAG repeat-targeting.
Response 7:
We thank the reviewer for this good suggestion. We added a sentence describing allele-specific silencing approaches and their references in the second paragraph of section 4 (page 9, line 413-415), in the revised manuscript.
Comment 8:
Issues of potential delivery of potential therapeutics to humans could be discussed.
Response 8:
We thank the reviewer for this comment. The delivery issue is discussed in the original manuscript in the second paragraph of the section 5 “Future Perspective” in page 10, line 447-448, describing that delivery methods that would enable therapeutic compounds to translocate efficiently through BBB and deliver them inside brains should be developed as a top priority issue.
Comment 9:
Also it would be valuable to mention about clinical trials with humans ongoing, related to polyQ disease-modifying approaches.
Response 9:
We thank the reviewer for this comment. According to the reviewer’s suggestion, we added examples of clinical trials that have been performed to develop the disease-modifying therapies for the polyQ diseases, including antisense oligonucleotide (ASO) for HD and leuprorelin for SBMA in page 10, line 418-420, and in page 10, line 455-page 11, line 472 in the revised manuscript.

Round 2
Reviewer 1 Report
The authors have fully improved the quality of the present review. I accept the present form of the review.